# Structure Integrity Analysis Using Fluid–Structure Interaction at Hydropower Bottom Outlet Discharge

**Mohd Rashid Mohd Radzi** [1,2], **Mohd Hafiz Zawawi** [1,*], **Mohamad Aizat Abas** [3],
**Ahmad Zhafran Ahmad Mazlan** [3], **Mohd Remy Rozainy Mohd Arif Zainol** [4], **Nurul Husna Hassan** [1],
**Wan Norsyuhada Che Wan Zanial** [1], **Hayana Dullah** [1] and **Mohamad Anuar Kamaruddin** [5]

1   Department of Civil Engineering, College of Engineering, Universiti Tenaga Nasional,
    Kajang 43000, Selangor, Malaysia
2   Hydro Life Extension Program (HELP), Business Development (Asset) Unit, TNB Power Generation Division,
    Petaling Jaya 46050, Selangor, Malaysia
3   School of Mechanical Engineering, Engineering Campus, Universiti Sains Malaysia,
    Nibong Tebal 14300, Pulau Pinang, Malaysia
4   School of Civil Engineering, Engineering Campus, Universiti Sains Malaysia,
    Nibong Tebal 14300, Pulau Pinang, Malaysia
5   School of Industrial Technology, Universiti Sains Malaysia, Nibong Tebal 14300, Pulau Pinang, Malaysia
*   Correspondence: mhafiz@uniten.edu.my

**Abstract:** Dam reliability analysis is performed to determine the structural integrity of dams and, hence, to prevent dam failure. The Chenderoh Dam structure is divided into five parts: the left bank, right bank, spillway, intake section, and bottom outlet, with each element performing standalone functions to maintain the overall Dam's continuous operation. This study presents a numerical reliability analysis of water dam reservoir banks using fluid–structure interaction (FSI) simulation of the bottom outlet structures operated at different discharge conditions. Three-dimensional computer-aided drawings were used to view the overall Chenderoh Dam. Next, a two-way fluid–structure interaction (FSI) model was developed to explore the influence of fluid flow and structural deformation on dam systems. The FSI modeling consists of Ansys Fluent and Ansys Structural modules to consider the boundary conditions separately. The reliability and performance of the reservoir bottom outlet structure was effectively simulated and recognised using FSI. The maximum stress on the bottom outlet section is 18.4 MPa, which is lower than the yield stress of mild steel of 370 MPa. Therefore, there will be no structural failure being observed on the bottom outlet section when the butterfly valve is fully closed. With a few exceptions, the FSI models projected that bottom outlet structures would be able to run under specified conditions without structural collapse or requiring interventions due to having lower stress than the material's yield strength.

**Keywords:** fluid-structural interaction; computational fluid dynamics (CFD); fluid flow dynamic; bottom outlet; Ansys

## 1. Introduction

Dams are hydraulic structures that are used to store water in reservoirs, pool water for agriculture, provide bed control, or divert flow away from crumbling banks or into diversion channels for flood control [1]. As such, a dam is designed to withstand the forces exerted by both static and dynamic water loadings [2]. Furthermore, the dam must withstand deterioration, ageing, and stresses caused by weather extremes and vibrations for longer than its design life span [3]. Dams have long been a major part of society's infrastructure, contributing to socioeconomic development and wealth through hydroelectric generating and residential water supply, for example [4]. A dam project typically includes a water-retaining structure (dam), a water-releasing structure (spillway), a water-conveying structure (conduits), and other components (such as turbines, power plants, etc.) [5].

The focus of this study is the Chenderoh Dam, a hydraulic structure situated in Tasik Chenderoh, Kuala Kangsar District, Perak, Malaysia. The dam construction began in 1928 and was completed in 1930, which was later officiated on 28 June 1930. The purpose of the Chenderoh Dam is for the hydroelectric scheme for the lower Perak region. The Chenderoh Dam continues its operation to date, making it the oldest hydroelectric dam and power station in Malaysia.

The Chenderoh Dam was the first dam to be built in the river (between 1927 and 1930), some 52 km downstream from the Kenering Dam. This dam is an Ambursen-type concrete hollow buttress dam, with gated and ungated overflow spillway sections over the crest. The gated section is equipped with a lower sector gate and an upper radial gate, and the ungated spill section was provided in 1972, with flashboards up to an elevation of 60.45 MASL, and, subsequently, a Japanese crest was anchored to the spillway crest downstream of the flashboards. The maximum height of the dam is 23 m and the crest length is about 390 m, with an angled axis. At full supply level (FSL), the reservoir volume is 95 million m$^3$, and the surface area is 20.5 km$^2$. The main powerhouse, with three Francis turbines of 10 MW each, together with the switchyard, is located on the right abutment. In 1981, a fourth turbine (Boving propeller-type) was installed at the dam toe, adjacent to the gated section, using one of the previous bottom outlets as the intake. The dam and appurtenant structures have been refurbished and rehabilitated several times.

Generally, the Chenderoh Dam structure can be divided into five parts, namely, the left bank, right bank, spillway, intake section, and bottom outlet, with each part performing standalone functions to ensure the continuous operation of the overall Chenderoh Dam.

Both the left bank and right bank resist and withstand the upstream water in Chenderoh Lake. The spillway structure regulates the water level of the reservoir from the upstream to the downstream parts through the control of sector gates. The primary power generation is located in the intake section, which consists of turbines and penstocks. Moreover, the bottom outlet serves as the secondary power generator and releases water from the upstream to downstream.

The operating condition of the left bank and right bank are dependent on the upstream water level. For the spillway's water regulating system, there are four operating conditions in which the sector gate could be open at a height of 4 feet, 10 feet, 12 feet, and 16 feet, respectively, yielding different discharge rates.

As for the intake section, its operation is based on the opening of the head gate and wicket gate. Under normal circumstances, both the head gate and wicket gate are either fully closed or fully opened. Nonetheless, under special conditions when required, the head gate could be fully opened or half-opened, while the wicket gate remains fully closed. Lastly, the operation of the bottom outlet is based on the opening condition of the butterfly gates, either fully closed or fully opened.

Computational fluid dynamics (CFD) focuses on computational transport phenomena, such as computational fluid dynamics, mass transfer, and heat transfer, as well as any other process that involves transportation phenomena [6,7]. CFD is useful in a wide variety of applications, for example, meteorological events, environmental risks, and the interaction of numerous objects with the air or water environment, to name a few. Numerical experiments can be carried out in a virtual flow laboratory [8]. Nowadays, the majority of CFD software programs have advanced to the point where they can simulate some types of fluid–structure interaction (FSI) applications [9]. CFD is research that uses computer technology to combine all of the equations in fluid flow (Richter, 2012), and it is a sophisticated numerical approach that is used in conjunction with physical modeling to model hydraulic processes [10]. Due to the general rapid growth of computer technologies, CFD has received more attention in recent years [11].

Nowadays, with the use of high-performance computers and more efficient algorithms, CFD can be a viable choice for making any analysis or experimentation easier, particularly when multiple prototypes are required throughout the design and testing process, as well as saving time and money [12,13]. The use of commercial CFD software to estimate time-averaged media velocity and pressure distributions along workpiece surfaces is now possible, thanks to the development of a rigorous approach for generating continuum media flow equations [14]. In addition, CFD solutions can describe the dynamic interaction between fluid flow, wind turbines, and floating platforms, allowing for full-scale simulations [12].

These equations show how a flowing fluid's velocity, pressure, temperature, and density are connected. Humans will be able to grasp and solve the Navier–Stokes equation more easily because it is analytical. Both compressible and incompressible fluids can be treated with these equations [13]. Understanding the physical events that occur in the flow of fluids around and within the chosen item or structure is the ultimate goal of CFD [15,16].

The Navier–Stokes equation is a partial differential equation (PDE) in fluid mechanics that describes the flow of incompressible fluids [16]. It is a term that defines the motion of viscous fluids [15]. Differential equations represent the link between the flow variables and their evolution in space and time in fluid flow equations [17]. The dynamic equilibrium of a fluid element can be used to derive the Navier–Stokes equations. To answer this problem using a computer, it must be converted to a discretized form [18]. Numerical discretization methods, such as the finite difference method (FDM), finite element method (FEM), and finite volume method (FVM), are used as translators (FVM) [19,20]. As a result, because discretization is dependent on them, the entire domain problem must be broken into several little portions [13]. The governing equations of CFD are the Navier–Stokes equations. The continuity equation, momentum equation, and energy equation can be obtained using mass, momentum, and energy conservation [20].

The exchange of energy between moving fluid and solid structures causes fluid–structure interaction (FSI) [15]. A fluid–structure interaction (FSI) is a multidimensional physics interaction between the rules of fluid dynamics and structural mechanics [21]. Fluid–structure coupling can occur in a variety of engineering domains, and it is considered critical in the design of many engineering systems [22]. In the case of FSI, a fluid and structural problem can be solved in conjunction with boundary conditions, described as a connected part of the boundary [21]. In order to compute the numerical solution, the strongly coupled equations of both problems must be solved simultaneously. Usually, FSI problems have a strong dependency between fluid and structure [23]. An FSI problem can be approached in one of two ways: monolithic or partitioned [22]. The monolithic technique involves solving the flow equations and structural equations at the same time, allowing for consideration of their mutual influence throughout the solution process [24,25].

The flow equations and the structure equations are solved independently in a partitioned FSI simulation, which means that the flow does not change while the structural equations are solved, and vice versa [25]. As a result, the partitioned approach necessitates the use of a coupling algorithm to incorporate the fluid–solid interaction into the system [26]. The partitioned approach, on the other hand, keeps software modularity and diversity [22]. For flow equations and structural equations, more efficient solution approaches are likely to be applied [27].

CFD models and other numerical models are increasingly being employed in engineering investigations [28]. Benchmark testing is commonly used to determine the validity of these models [29]. This is done to quantify the agreement between the model's predictions and the real world, which is represented by observations in experiments [30]. This approach implies that all real-world variables important to the investigation are adequately measured in the experiments and in the model's predictions [31].

Structural analysis is important for a hydraulic structure. A hydraulic structure needs to be analyzed to ensure its stability and integrity. Structural analysis is a method of analyzing a structural system in order to predict its behavior and consequences using mathematical equations and physical laws [19]. All structures that must withstand varying loads are subject to this type of study. Structural analysis computes a structure's deformations, internal forces, stresses, support responses, accelerations, and stability using applied materials science, mechanics, and applied mathematics [32]. The analysis findings are utilized to confirm the structure's strength and usability [33].

Stress is a physical quantity that expresses the internal forces that contiguous particles of a continuous material exert on each other in continuum mechanics, whereas strain is the measure of the material's deformation [33]. The force per unit area applied to a material is referred to as stress [19].

Failure due to structural weakness is referred to as structural uncertainty [19]. Hydraulic structure failures can be caused by soil saturation and instability, erosion, hydraulic soil failures, wave action, hydraulic overloading, structural collapse, material failure, and so on [34–36]. Hydraulic erosion, high pore-water pressure, seismic stresses, and other variables all contribute to embankment failures [34].

Every construction should be designed with environmental, ecological, and public safety in mind [36]. Hydraulic structures have diverse characteristics, such as shape and size, depending on the project. This is dependent on the discharge and the function to be carried out properly [33]. Hydraulic physical modeling or CFD modeling may be useful for the design of unique structures that do not meet the guidelines offered [9]. In the case of FSI problems, the structure's equations should be written in such a way that substantial deformations of the structure are unlikely [37].

The main objective of this paper is to study the fluid–structure interactions of the bottom outlet structures of Chenderoh Dam operated at different discharge conditions.

## 2. Methodology

### 2.1. Three-Dimensional Computer-Aided Drawings of Chenderoh Dam

The overall Chenderoh Dam consists of five sections, namely, the intake, right bank, sector gate, bottom outlet, and left bank. A top view of the overall Chenderoh dam is shown in Figure 1, based on the provided drawing plans, which can be segregated into the structures of intake, right bank, sector gate, bottom outlet, and left bank. Three-dimensional drawings of the overall dam were generated using SolidWorks 2017 software, based on the build drawing plans and dimensions provided by the TNB Chenderoh Hydropower Station. Figures 2 and 3 exhibit the entire three-dimensional sketch of the Chenderoh Dam from downstream and upstream perspectives, respectively.

Figure 4 depicts the detailed cross-sectional views of the bottom outlet: (a) downstream/front view of the bottom outlet including the turbine house, penstock, etc.; (b) top view of the bottom outlet including the turbine house, penstock, etc.; (c) backside of the bottom outlet (foundation, trash rack, etc.); (d) tented gate connection to the penstock, butterfly gate no. 1 and turbine, and (e) penstock and butterfly gate no. 2. The bottom outlet is another important feature of the dam structure since it is the second location for the source of power generation for the station and the location for the release of water from the upstream to the downstream. For the bottom outlet, the propeller-type turbine is used for the power generation located inside the turbine house of the dam. The release of water at the bottom outlet location is controlled by two butterfly gates, which are separately shown in Figure 4d,e. These gates are essential since there will be another case study of water surging inside the penstock, and the effects of flow-induced vibration to the whole bottom outlet part.

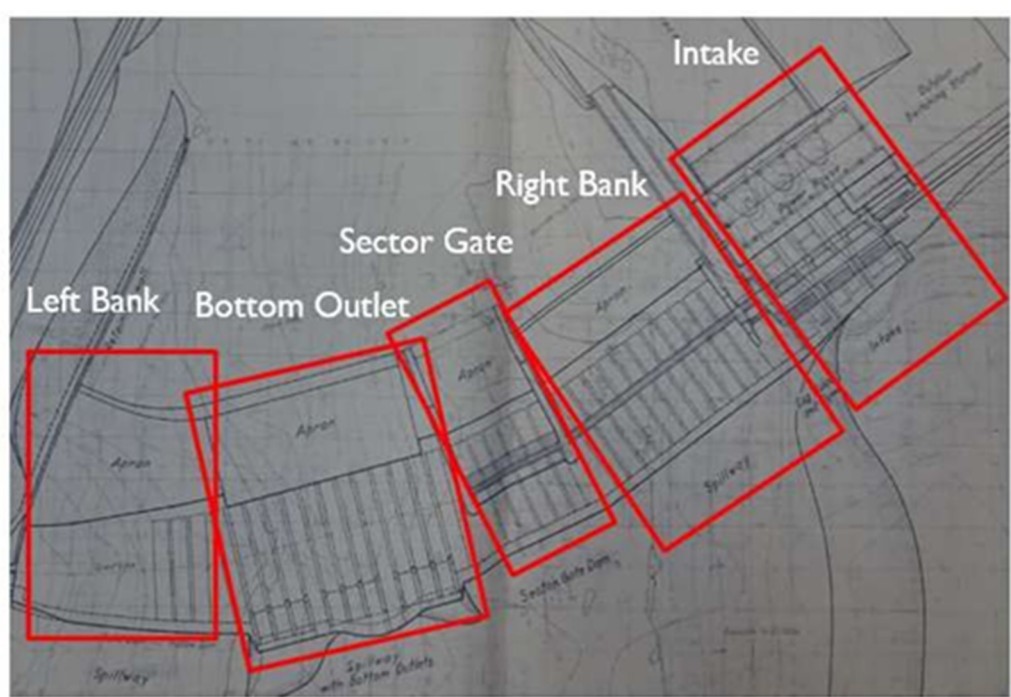

**Figure 1.** Top view of the overall Chenderoh Dam based on the provided drawing plans.

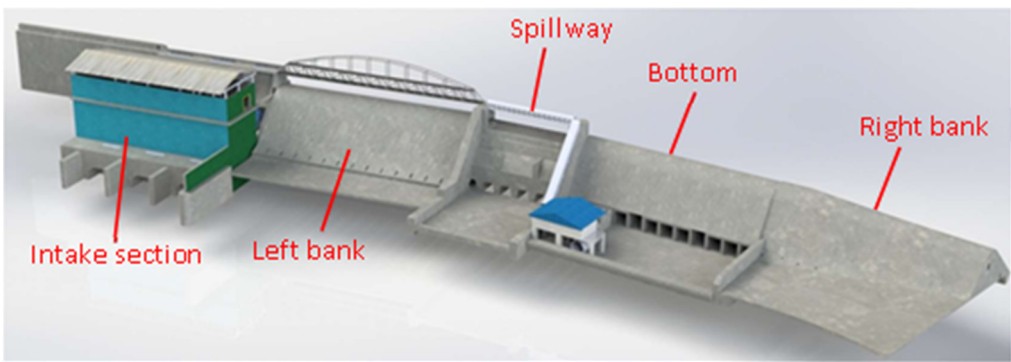

**Figure 2.** Downstream view of the overall Chenderoh Dam.

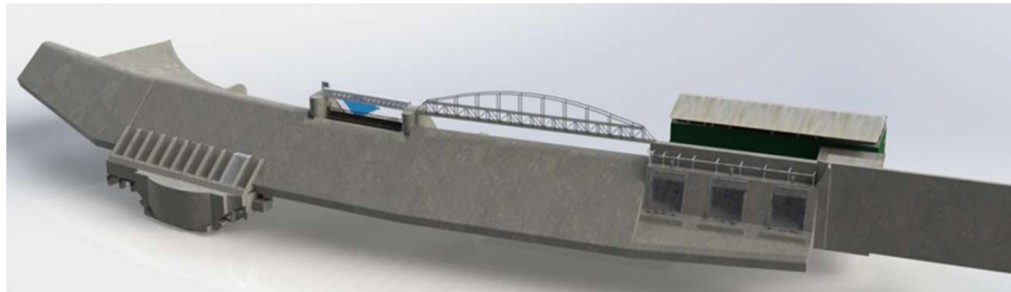

**Figure 3.** Upstream view of the overall Chenderoh Dam.

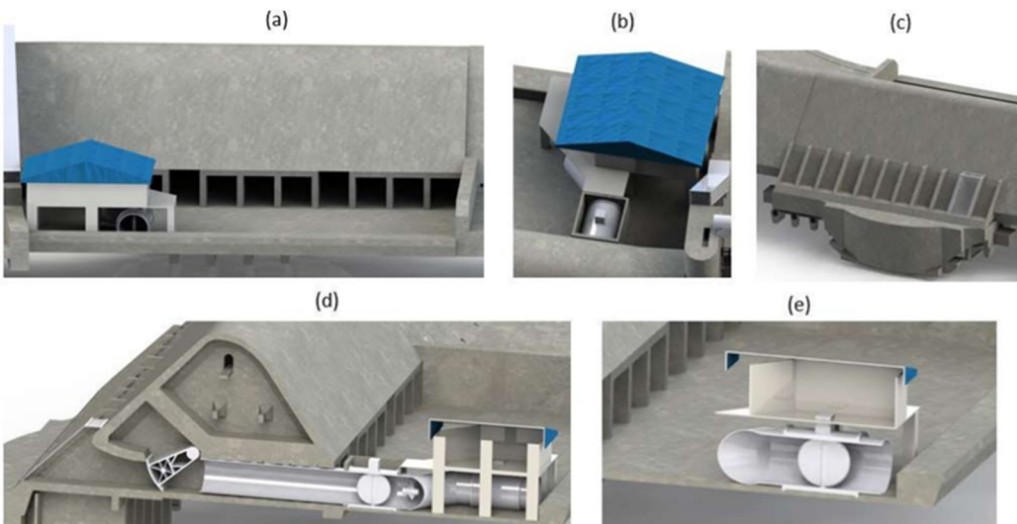

**Figure 4.** Detailed cross-sectional views of bottom outlet. (**a**) Downstream/front view of bottom outlet including the turbine house, penstock, etc.; (**b**) top view of bottom outlet including the turbine house, penstock, etc.; (**c**) backside of bottom outlet (foundation, trash rack, etc.); (**d**) tented gate connection to the penstock, butterfly gate no. 1, and turbine, and (**e**) penstock and butterfly gate no. 2.

*2.2. Fluid–Structure Interaction Numerical Simulation*

The two-way fluid–structure interaction (FSI) was developed to investigate the coupling impact of fluid flow and structural deformation in dam systems. The flow chart in Figure 5 summarises the general sequences involved in the current FSI modelling, from the domain and mesh generation to the boundary condition assignment. Ansys Fluent and Ansys Structural modules were used to investigate the fluid and structural domains separately.

Both the fluid domain and the structural domain were investigated independently using the Ansys Fluent and Ansys Structural modules, respectively. The fluid flow in the present simulation is a rectangular enclosure surrounding the bottom outlet structure, which is the location where the fluids are predicted to occur, as shown in Figure 6. The Navier–Stokes equations, which include a momentum equation and a continuity equation, regulate the fluid phase. The linear elastic equation guides the structural phase.

The numerical fluid and structural domains are then discretized via mesh generation in the next stage. Using the improved mesh size, structural hexagonal meshes were created on both the mesh and fluid domains. The dynamic mesh on the fluid flow is enabled in the present FSI simulation to account for structural domain movement and deformation.

A multiphase volume of fluid model with an implicit scheme regulated by the transport equation was used to follow the flow front of the water. Furthermore, the k-model simulates water flow turbulence. First order upwind, least square cell method, and SIMPLE pressure-velocity coupling are the solutions employed in this paper. Following that, the Ansys Fluent and Ansys Mechanical settings were connected using the system coupling module to provide two-way data transmission.

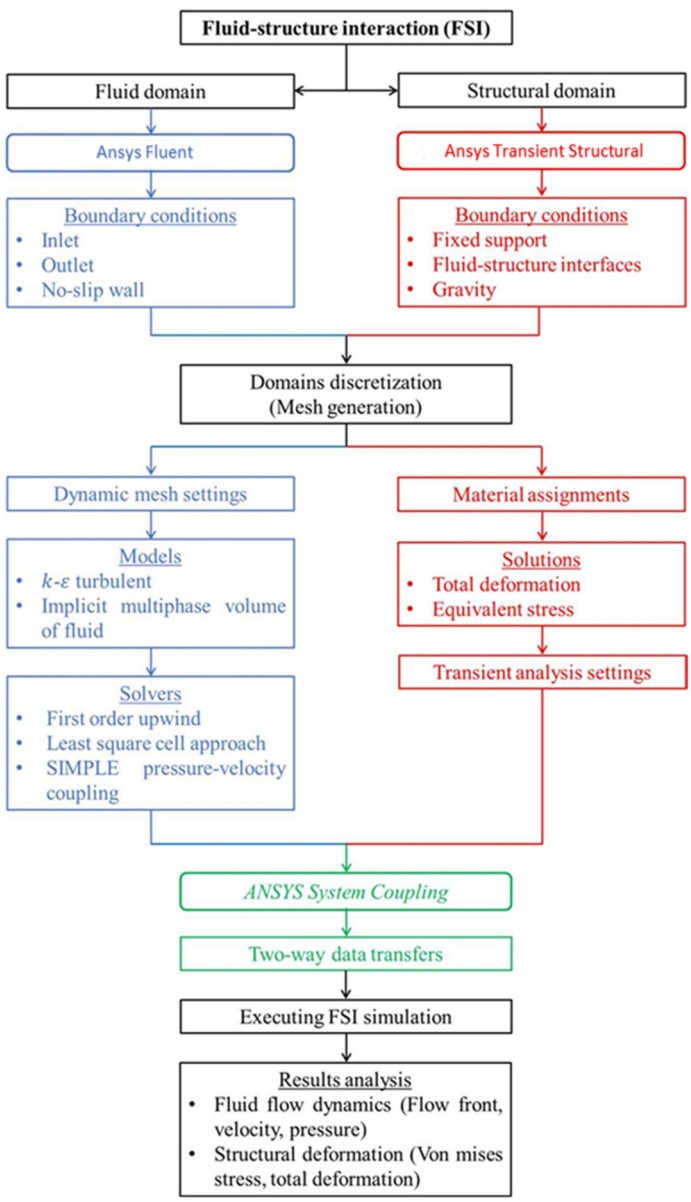

**Figure 5.** Flow sequences of the fluid–structure interaction numerical simulation.

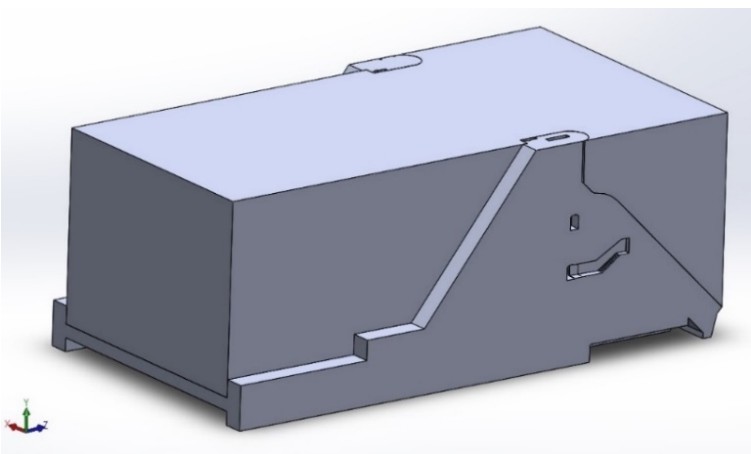

**Figure 6.** Schematics of fluid and structure domain.

2.2.1. Governing Equations

In the Ansys Fluent module, the current flow simulation was governed by the incompressible and isothermal Navier–Stokes equations, with the respective governing continuity and momentum equations as follows:

$$\frac{\partial \rho}{\partial t} + \nabla \cdot \left( \rho \vec{v} \right) = 0 \tag{1}$$

$$\frac{\partial \rho}{\partial t} \left( \rho \vec{v} \right) + \nabla \cdot \left( \rho \vec{v} \vec{v} \right) = -\nabla p + \nabla \cdot \bar{\bar{\tau}} + \rho \vec{g} + \vec{F} \tag{2}$$

where $p$ is the static pressure; $\rho$ is the density; $\vec{g}$ denotes the gravitational acceleration; $\vec{F}$ is the external body force; and $\bar{\bar{\tau}}$ is the stress tensor.

The standard k-$\varepsilon$ turbulence model was employed in the current simulation, which solves the following transport equations:

$$\frac{\partial}{\partial t} \left( \rho k \right) + \frac{\partial}{\partial x_i} \left( \rho k u_i \right) = \frac{\partial}{\partial x_j} \left[ \left( \mu + \frac{\mu_t}{\sigma_k} \right) \frac{\partial k}{\partial x_j} \right] + G_k + G_b - \rho \varepsilon - Y_M + S_k \tag{3}$$

$$\frac{\partial}{\partial t} \left( \rho \varepsilon \right) + \frac{\partial}{\partial x_i} \left( \rho \varepsilon u_i \right) = \frac{\partial}{\partial x_j} \left[ \left( \mu + \frac{\mu_t}{\sigma_\varepsilon} \right) \frac{\partial \varepsilon}{\partial x_j} \right] + C_{1\varepsilon} \frac{\varepsilon}{k} \left( G_k + G_{3\varepsilon} G_b \right) - C_{2\varepsilon} \frac{\varepsilon^2}{k} + S_\varepsilon \tag{4}$$

where $G_k$ is the generation of turbulence kinetic energy due to the mean velocity gradients; $G_b$ is the generation of turbulence kinetic energy due to buoyancy; $Y_M$ is the contribution of the fluctuating dilatation in compressible turbulence to the overall dissipation rate; $\sigma$ denotes the turbulent Prandtl number; $S$ is the user-defined source term; and the constant terms are $C_{1\varepsilon}$ and $C_{2\varepsilon}$.

To track the water–air interface, the multiphase volume of fluid (VOF) model was employed, which is governed by the transport equation:

$$\frac{\partial \alpha}{\partial t} + \vec{V} \bullet \nabla \alpha = 0 \tag{5}$$

Numerical software packages solve problems using a series of discrete points. Each point, or node, adds a degree of freedom (DOF) to the system. Therefore, the more DOFs in the model the better it will capture the structural behaviour. Each DOF adds complexity and increases solving time. The simulation needs to balance the complexity of the model with the solving time. Too few DOFs, and the response could be incorrect, and too many DOFs, and the model could take days to run. The Ansys Mechanical module was used to run the simulation on the structural domains. The findings of the static structural analysis for the dam gates were fed as input, after the pressure loads were obtained from the fluid simulations. The following equation was used by the Ansys Mechanical APDL for the static linear analyses.

$$[K]\{u\} = \{F^a\} + \{F^r\} \tag{6}$$

where $\{u\}$ is the nodal degree of freedom (DOF) vector, and $[K]$ is the total stiffness or conductivity matrix, which defined as

$$[K] = \sum_{m=1}^{N} [Ke] \tag{7}$$

with the number of elements, $N$, and the element stiffness or conductivity matrix, $[Ke]$. On the right-hand side of equation (7), $\{F^r\}$ is the nodal reaction load vector, and $\{F^a\}$ is the total

applied load vector, which is the sum of the applied nodal load vector, $\{F^{nd}\}$, and the total of all the element load vector effects (pressure, acceleration, thermal, gravity), and $\{F^e\}$:

$$\{F^a\} = \left\{F^{nd}\right\} + \{F^e\} \tag{8}$$

Ansys Mechanical uses finite element (FE) techniques on all structural models to construct a system of simultaneous linear equations, as detailed in the previous section. After that, either a direct elimination technique or an iterative method is used to solve the equations. The preconditioned conjugate gradient (PCG) solver was used in this simulation as well. General CG methods cast the solution in the form of a series of vectors, $\{p_i\}$, to solve standard systems of equations in the form of previous equations recursively.

$$\{u\} = \alpha_1\{p_1\} + \alpha_2\{p_2\} + \alpha_3\{p_3\} + \ldots + \alpha_m\{p_m\} \tag{9}$$

2.2.2. Boundary Conditions

The setup of the numerical simulation was initiated with the assignation of suitable boundary conditions on the fluid domain and structural domain. Generally, the boundary conditions imposed on the fluid domain are no-slip wall, inlet and outlet, while, for the structural domain they are fixed support, gravity, and fluid structure interface. The velocity inlet was assigned to the reservoir inlet at one end of the fluid domain, while the pressure outlet was assigned to the opposite end. The water would constantly flow in the fluid domain at the inlet surface with a constant velocity of 1 m/s at the reservoir intake, with a water level height of 44.8 m.

Furthermore, the entire fluid domain was adjusted to a 0 Pa atmospheric pressure state (gauge). The no-slip boundary condition was enforced together with the fluid-structure interface condition on all the contacting surfaces between the fluid and structure domains. Furthermore, the dam spillway's foundation was designated as a fixed support. There is also gravity of a magnitude of 9.81 m/s$^2$ acting downward enabled on both domains.

2.2.3. Mesh

The grid-generation procedure was then used to discretize both the fluid and structural domains into tiny elements. Meshes are divided into two categories: structured and unstructured grids. In structured grids, the cells are ordered and numbered according to indices, such as I, j, and k. Planar cells with four edges (2-D) or volumetric cells with six faces make up a structured grid (3-D). Although the cells are ordered according to indices, they might be geometrically deformed. Unstructured grid cells, on the other hand, cannot be uniquely identified by indices, and the relationship between adjacent cells must be accounted for by other means, which usually involves another set of memory storage. To overcome this deficiency, the grid density was changed so that an extremely tiny mesh was allotted at the necklace vortices, detached shear layers, and near-wake zone areas [38]. Furthermore, the cells come in a variety of shapes, although the most common are triangles or quadrilaterals (2-D), and tetrahedrons or hexahedrons (3-D).

Unstructured grids are often used for complex geometries because they are easier to design by the user with fewer tedious grid generation processes, are more robust, and can fit steep angles in the geometry without compromising grid skewness. Structured grids, on the other hand, are preferable in other ways. For example, with a structured grid, fewer cells are normally generated than with an unstructured grid.

Furthermore, for the same number of cells, structured grids allow better control over local grid refinement and sharper resolution in border layers than unstructured grids. Calomino et al. [39] divided the domain into pieces depending on direction using the basic geometric decomposition technique [39]. In this method, the domain is divided into segments by direction [40]. Due to the generally huge space for the water dam geometry and the need to replicate complicated geometry, the meshes were generated using an unstructured grid. This is mostly owing to the FSI interactions, which occur frequently in

complex geometries. Combining tetrahedrons and hexahedrons mesh types is also possible, thanks to the unstructured mesh allowances. As a result, during the FSI simulation, better findings will be captured.

To determine the optimum mesh types for numerical modal analysis, a mesh independent analysis was conducted. Table 1 shows the number of elements, with their corresponding curvature and proximity minimum size, for the fluid mesh of the bottom outlets. The mesh with the curvature and proximity minimum size of 0.55 m produced the highest maximum velocity of 124.1 m/s. It can be observed in Figure 7 that the maximum velocity value starts to saturate at a 0.45 m mesh size. This indicates that this size is the optimum mesh size, as using the smaller mesh size will not affect the results greatly. The generated mesh models of the dam structures to be used in the numerical modal analysis are shown in Figure 8.

**Table 1.** Comparison between mesh size, number of elements, and maximum velocity for the fluid domain of the intake section.

| Mesh Size (m) | Element Number | Maximum Velocity (m/s) |
|:---:|:---:|:---:|
| 0.55 | 386,358 | 124.1 |
| 0.5 | 452,317 | 122.467 |
| 0.45 | 534,276 | 122.367 |
| 0.35 | 797,630 | 122.332 |
| 0.3 | 1,027,179 | 122.3 |

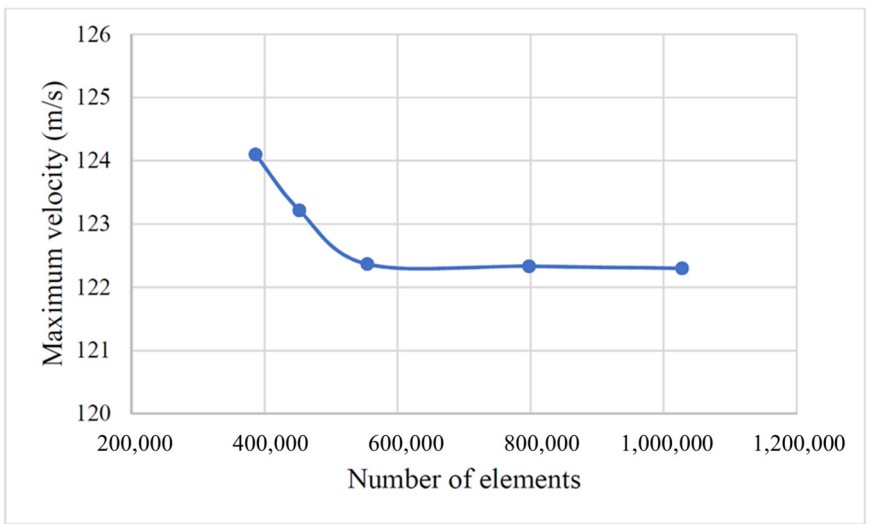

**Figure 7.** Mesh sensitivity analysis of five different mesh sizes for the fluid domain.

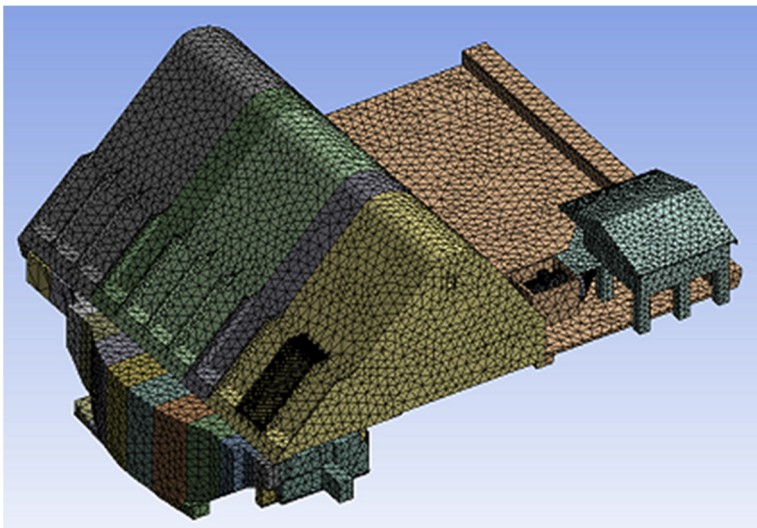

**Figure 8.** Meshing picture of bottom outlet 2.2.4 Ansys fluid flow (Fluent).

Referring to the second objective of this study, which is to analyze the fluid dynamics from the reservoir to the sector gate spillway of Chenderoh Dam at different sector gate openings, the calculations of the model are performed using the robust CFD-solver Fluent. Fluent provides various multiphase models that are based in the Eulerian–Eulerian approach [41]. Although a Eulerian model that has been selected for the numerical approach may require more computational effort, it can handle a wider range of particulate loading values and is more accurate than the other available multiphase models in Fluent [42].

In this multiphase model, the different phases are treated mathematically as interpenetrating continuous, and therefore the concept of phasic volume fraction is introduced, where the volume fraction of each phase is assumed to be a continuous function of space and time. The sum of the volume fractions of the various phases is equal to unity. An accordingly modified set of momentum and continuity equations for each phase is solved. Pressure and inter-phase exchange coefficients are used in order to achieve coupling for these equations [42].

In Ansys Fluent, pressure and inter-phase exchange coefficients are used to achieve coupling between the different equations used to model fluid flow and inter-phase interactions. The pressure coefficient is used to couple the Navier–Stokes equations, which describe the fluid flow, with the pressure equation, which enforces mass conservation. The pressure coefficient is a numerical parameter that relates the change in pressure to the change in the velocity of the fluid. It is used to ensure that the velocity and pressure fields are consistent with each other. The inter-phase exchange coefficient is used to couple the different phases in a multiphase flow simulation, such as gas and liquid. It represents the rate at which mass, momentum, and energy are exchanged between the phases, and it is used to ensure that the different phases are in thermal and mechanical equilibrium. The inter-phase exchange coefficient is dependent on the physical properties of the phases, as well as the geometry of the system being simulated.

The type of solver used in this simulation is the SIMPLE (Semi-Implicit Method for Pressure-Linked Equations) family of algorithms used for introducing pressure into the continuity equation. The SIMPLE algorithm uses a relationship between velocity and pressure corrections to enforce mass conservation and to obtain the pressure field. If the momentum equation is solved with a guessed pressure field, $p^*$, the resulting face flux, $J_f^*$, is computed from Equation (10).

$$\sum_f^{N_{faces}} J_f A_f = 0 \tag{10}$$

$$J_f = \hat{J}_f + d_f(p_{c0} - p_{c1}) \tag{11}$$

where $J_f$ = the face flux; $\hat{J}_f$ = contains the influence of velocities in these cells; $d_f$ = the function of $\tilde{a}_P$; the average of the momentum equation = $a_P$; the coefficient for the cells on either side of face f; and $p_{c0}$ and $p_{c1}$ = the pressure between two cells on either side of the face.

$$J_f^* = \hat{J}_f^* + d_f(p_{c0}^* - p_{c1}^*) \tag{12}$$

does not satisfy the continuity equation. Consequently, a correction, $J_f'$, is added to the face flux, $J_f^*$, so that the corrected face flux

$$J_f = J_f^* + J_f' \tag{13}$$

satisfies the continuity equation. The SIMPLE algorithm postulates that $J_f'$ be written as

$$a_P p' = \sum_{nb} a_{nb} p_{nb}' + b \tag{14}$$

where the source term, $b$, is the net flow rate into the cell:

$$b = \sum_f^{N_{faces}} J_f^* A_f \tag{15}$$

The pressure-correction equation (Equation (6)) may be solved using the algebraic multigrid (AMG) method. Once a solution is obtained, the cell pressure and the face flux are corrected using:

$$p = p^* + \alpha_p p' \tag{16}$$

$$J_f = J_f^* + d_f(p_{c0}' - p_{c1}') \tag{17}$$

Here, $\alpha_p$ is the under-relaxation factor for pressure. The corrected face flux, $J_f$, satisfies the discrete continuity equation identically during each iteration.

### 2.2.4. Ansys Transient Structural

This type of analysis is used to determine the dynamic response of a structure under the action of any general time-dependent loads. It could be used to determine the time-varying displacements, strains, stresses, and forces in a structure as it responds to any transient loads [43]. The time scale of the loading is such that the inertia or damping effects are considered to be important. It also could be used to examine deflections, deformations, stresses, and strains on assemblies, part-by-part or at the feature level [15]. The input parameters for a few properties, such as the density of the structural concrete or Young's modulus for structural steel, were set in the engineering data before the simulation was run.

### 2.2.5. Ansys System Coupling

After the transient structural and fluid flow have been analyzed, system coupling was then run to analyze the FSI. System coupling is a process where the interpretations are made from the solutions given by the numerical models. Then, the relationship between the patterns of flow and input parameters or structure may be concluded.

## 3. Results and Discussion

### 3.1. Fluid–Structure Interaction Numerical Simulation on Bottom Outlet Structures

The performance and reliability of Chenderoh Dam's structures were assessed quantitatively based on the findings attained from the numerical FSI simulations, in terms of the flow dynamics and structural associated parameters. This is to ensure the bottom outlet structures can withstand the enormous pressures from the rapid and continuous water flow from the reservoir dam, without losing their structural integrity, ultimately, to eliminate the risk of dam failures. The bottom outlet is based on the open condition of a butterfly valve opening.

The investigated operating conditions for the bottom outlet section are based on the opening and closing the butterfly valve. The sudden closing of the butterfly valve is expected to cause minor surging to occur, which might lead to slight vibrations and a high region of stresses. It should be noted that the drawings for the butterfly valve and penstock pipes are not provided by the hydropower station and TNB. Therefore, the current numerical findings presented are based on the assumed geometry and dimensions of the penstock pipe and butterfly valve, and the material properties of concrete with Young's modulus of 30 GPa and mild steel of 210 GPa.

Figure 9 shows the condition of the butterfly valve. When the butterfly valve is fully opened, the face of the butterfly valve is parallel to the water flow. When the butterfly valve is fully closed, the face of butterfly valve is perpendicular to the water flow.

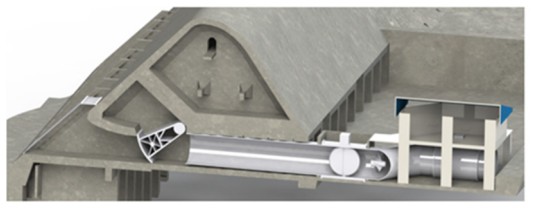
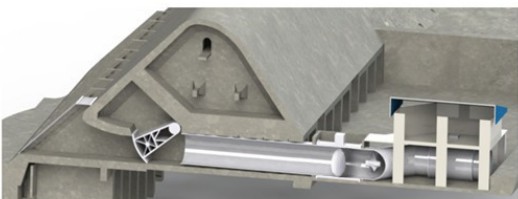

(**a**) Fully opened                                          (**b**) Fully closed

**Figure 9.** Condition of butterfly valve when fully opened and fully closed.

Figure 10 presents the numerical contours of the bottom outlet section when the butterfly valve is fully opened. A high-velocity region is located on the penstock pipe, with values in the ranges of 14.1–9.4 m/s, due to the water flow in the confined, narrow penstock pipe. Such a rapid flow will induce a large force on the inlet section and internal structures of the penstock pipes. Beyond the penstock region, the water velocity subsided to a value of 4.7 m/s. A hydrostatic pressure contour was observed on the upstream. A high deformation was observed on the region of the concrete wall in the downstream section, with values of 11.9–17.9 mm. As the higher deformation was located on the outer fin of the concrete wall, care must be taken to monitor the relevant structure for potential failure and crack. There is a high-stress region of about 73.9 MPa located on the inner section of the penstock. Since the maximum stress of the penstock is less than the yield stress of mild steel of 370 MPa, no structural failure will be observed.

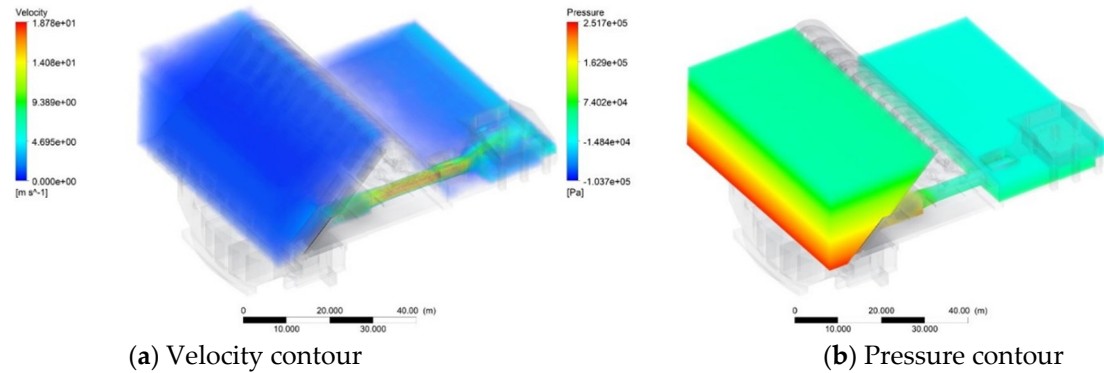

(**a**) Velocity contour                                          (**b**) Pressure contour

**Figure 10.** *Cont.*

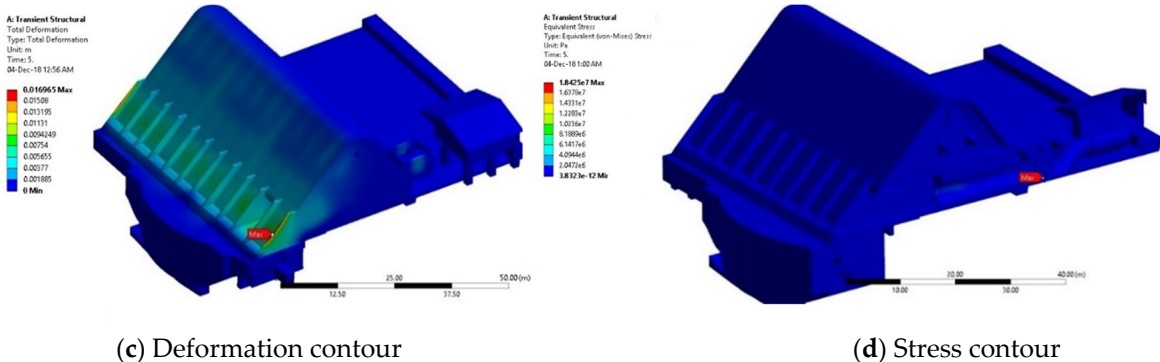

(**c**) Deformation contour        (**d**) Stress contour

**Figure 10.** Contours on the bottom outlet with fully opened butterfly valve.

Figure 11 presents the numerical contours of the bottom outlet section when the butterfly valve is fully closed. A high-velocity region is observed in the penstock pipes, with values in the range of 19.97–13.32 m/s, which later subsides to a value of 6.66 m/s beyond the penstock region. The high-deformation region is observed at the concrete wall of the downstream section, in the range of 13–17 mm. These deformations are caused by the nearly stagnant upstream water and are expected to have minimal impact to the structure. Furthermore, the maximum stress on the bottom outlet section is 18.4 MPa, which is lower than the yield stress of mild steel of 370 MPa. Therefore, there will be no structural failure observed on the bottom outlet section when the butterfly valve is fully closed.

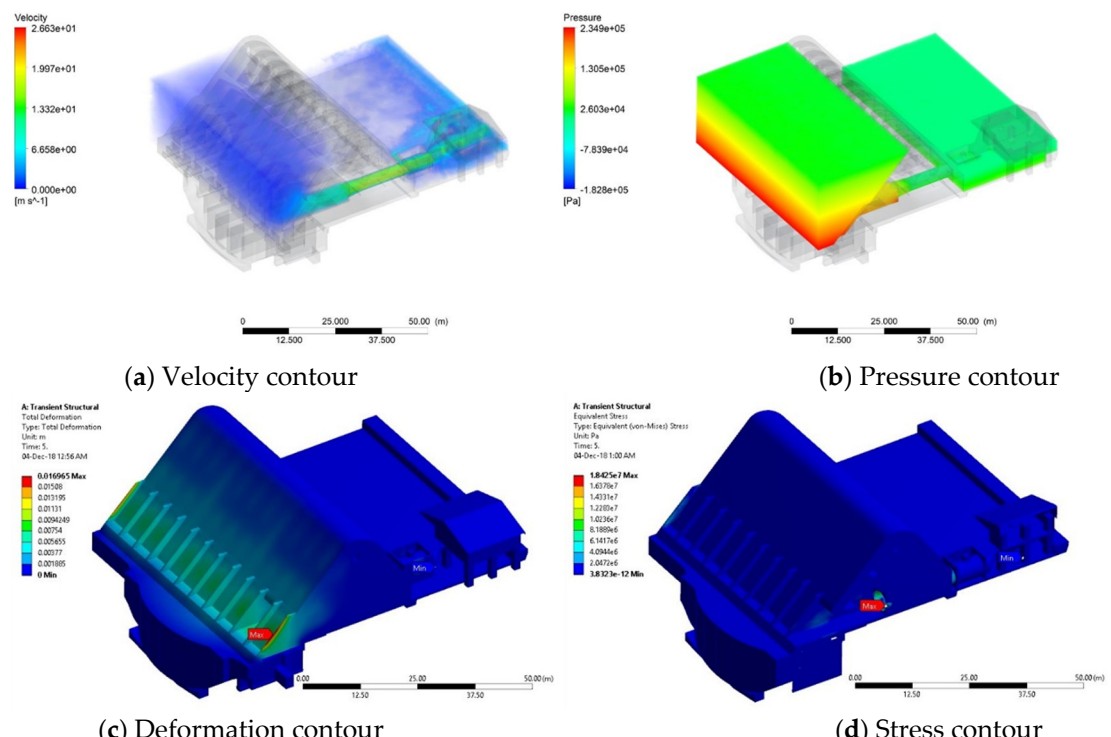

(**a**) Velocity contour        (**b**) Pressure contour

(**c**) Deformation contour        (**d**) Stress contour

**Figure 11.** Contours on the bottom outlet with fully closed butterfly valve.

### 3.2. Structural Analysis on Bottom Outlet Structures

In Figure 12, the bottom outlet operates at Case 1. Figures 12 and 13 show the stress contours on the bottom outlet when the butterfly valve is fully opened (Case 1) and fully closed (Case 2), respectively. The bottom outlet comprises the penstock adjoining two butterfly gate valves. The maximum stresses of 73.92 MPa occur at the gate valve during 100% opening of the gate valve. Meanwhile, a maximum stress value of 18.425 MPa, when the gate valve is closed, occurs at the penstock surface intake. Since the maximum stress of

the penstock is less than the yield stress of mild steel of 370 MPa, no structural failure will be observed.

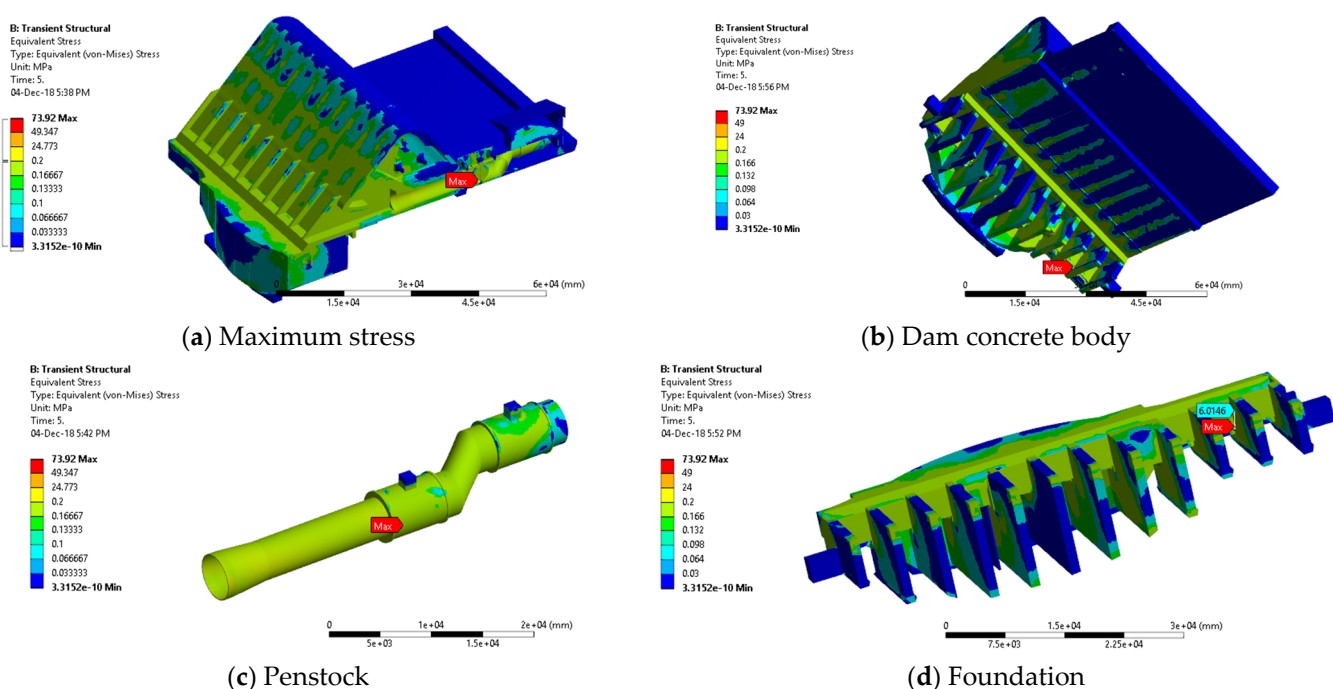

(**a**) Maximum stress   (**b**) Dam concrete body

(**c**) Penstock   (**d**) Foundation

**Figure 12.** Bottom outlet operates at Case 1.

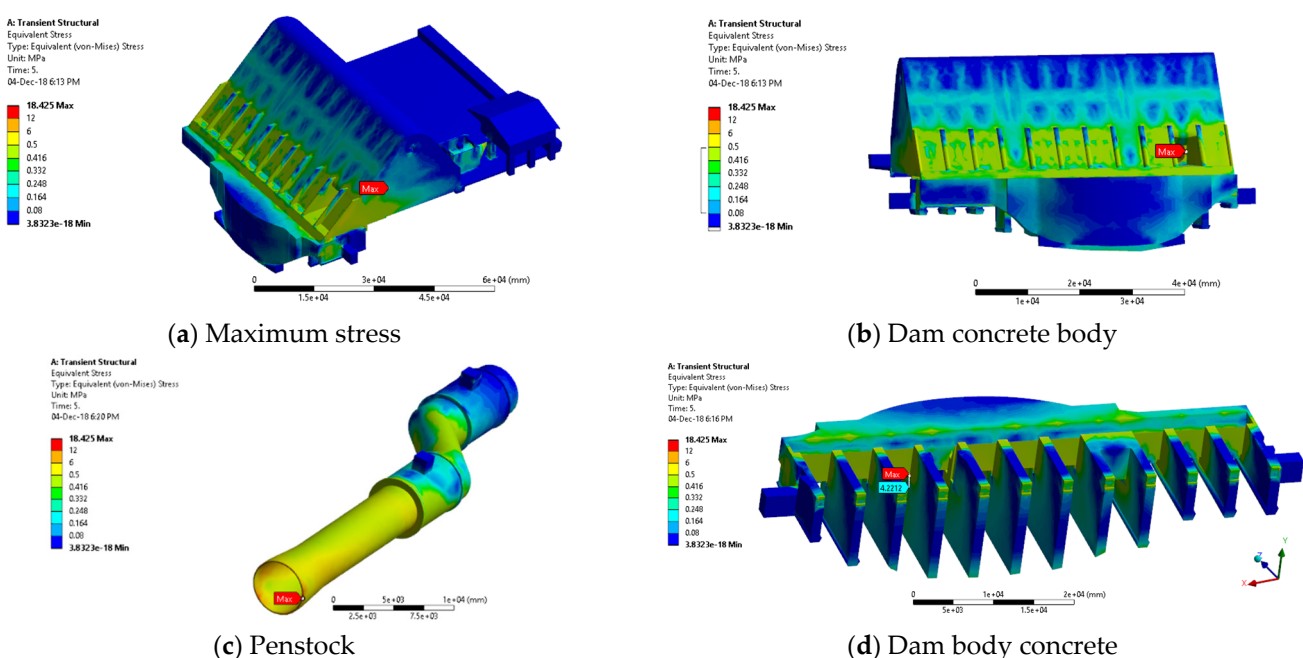

(**a**) Maximum stress   (**b**) Dam concrete body

(**c**) Penstock   (**d**) Dam body concrete

**Figure 13.** Bottom outlet operates at Case 2.

These stresses are related to the flow velocity at the bottom outlet, by which greater velocity will produce maximum stresses to the penstock surface intake. This is proven by Manafpour and Rovesht, 2017 in their study, where the flow velocity increased with a smaller gate opening, but it gradually increases with the gate closure, as shown in Table 2 [44]. Hence, the greater the velocity, the greater the stress created on the penstock surface. The extraction of the data is conducted; therefore, the location of maximum stress

is found at the intake. The values at the mechanical equipment and stilling basin are also high.

**Table 2.** Comparison of study conducted with that of Manafpour and Rovesht, 2017.

| Case | Chenderoh Dam–Bottom | Case | Seymareh Dam [44] |
|---|---|---|---|
| Butterfly valve is fully closed | 14.1 | Gate opening–10% | 2.0 |
| Butterfly valve is fully opened | 19.97 | Gate opening–30% | 4.0 |
| - | - | Gate opening–70% | 8.0 |
| - | - | Gate opening–100% | 18.0 |

## 4. Conclusions

By using fluid-structure interaction (FSI), which utilized both Ansys Fluent and Ansys Transient Structural, both the fluid domain and structural domain aspects of the reservoir bank of a dam have been successfully investigated. The impact of the upstream water flow on the dam structures, alongside their interaction phenomena, were numerically simulated by a fluid–structure interaction (FSI) numerical approach. The model's computations in Ansys fluid flow were carried out with the help of the robust CFD-solver Fluent, whereas Ansys Transient Structural analysis was used to determine the dynamic response of the structure under the influence of any time-dependent loads. After analyzing the transient structural and fluid flow, system coupling was performed to analyze the FSI. System coupling is a process in which interpretations are derived from numerical model results. The relationship between the flow patterns and the input parameters or structure may then be determined. To assess the reliability of the FSI simulations on the dam structures, a quantitative validation was conducted by comparing both the numerical and experimental volume flow rates at the penstocks at the intake sections.

From the FSI numerical simulation reports, it was assessed that the high-stress region located at the inner section of the penstock is about 73.9 MPa. The numerical contours of the bottom outlet section, when the butterfly valve was fully opened and fully closed, had a maximum stress on the penstock less than the yield stress of mild steel of 370 MPa, so no structural failure will be observed. Generally, the FSI simulations predicted that the bottom outlet structures will be able to operate under the prescribed conditions without structural failure or required interventions, due to having lower stress that the material's yield strength, with only a few exceptions. It is recommended for future studies to consider other parts of Chenderoh Dam to be analyzed, namely the left bank, right bank, spillway, and intake section to fully validate the Dam's reliability and performance.

**Author Contributions:** Writing—review and editing, M.R.M.R., M.H.Z., M.A.A., A.Z.A.M., M.R.R.M.A.Z., N.H.H., W.N.C.W.Z., H.D. and M.A.K. All authors have read and agreed to the published version of the manuscript.

**Funding:** This research was funded by UNITEN R&D Sdn. Bhd, U-TG-RD-21-19.

**Data Availability Statement:** Not applicable.

**Acknowledgments:** The authors would like to acknowledge Universiti Tenaga Nasional Berhad under UNITEN R&D Sdn. Bhd for providing the facilities and financial assistance of project code U-TG-RD-21-19.

**Conflicts of Interest:** The authors declare no conflict of interest.

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
