# Peer review of "Structure Integrity Analysis Using Fluid–Structure Interaction at Hydropower Bottom Outlet Discharge"

_water, doi:10.3390/w15061039_

Round 1

Reviewer 1 Report

please see attached document.

Reviewer 2 Report

The manuscript needs a major revision according to the attached file.

Author Response

Point 1: The 'Abstract' is quite short. Please add more about the novelty and findings of your study in detail.

Response 1: Thank you so much for your comments. The novelty and findings of this study has been added in abstract.

Point 2: Line 45: please remove "electric" after "hydroelectric".

Response 2: Thank you so much for your comments. The "electric" after "hydroelectric" has been removed.

Point 3: Line 101: I do not agree that CFD is based on the Navier-Stokes equations. Actually, other kinds of fluid dynamic equations can be used in CFD-based modeling’s.

Response 3: Thank you so much for your comments. The statement has been amended in this manuscript.

Point 4: The 'Introduction' section is long and includes general descriptions of CFD that many readers of this paper are aware of them. Please substitute those paragraphs with your specific case. Please focus on the problem that you aim to solve in this paper.

Response 4: Thank you so much for your comments. The introduction section has been amended in this manuscript.

Point 5: Lines 167 to 171: Is that section really related to your study? Please eliminate it if it is not directly relevant to your investigation.

Response 5: Thank you so much for your comments. The lines 167 to 171 are not related to this study. Hence the lines 167 to 171 has been removed.

Point 6: Lines 184 and 185: Please add a paragraph about the state-of-art of your study, as well as the novelty of your study with respect to the previous investigations. The 'Introduction' of your paper should be significantly revised.

Response 6: Thank you so much for your comments. The introduction section has been revised in this manuscript.

Point 7: Line 217: Please add description of subfigures (a), (b),... in the figure's caption.

Response 2: Thank you so much for your comments. The description of subfigures has been added.

Point 8: Line 219 to 224: You need to add a comprehensive description about the flowchart in Figure 5. Please explain each part in detail. That figure is ambiguous to many readers.

Response 8: Thank you so much for your comment. A comprehensive description about the flowchart in Figure 5 has been added.

Point 9: Figure 5 shows that k-epsilon turbulence model was employed in your study. Actually, that model is outdated and several more accurate models have already developed and recommended, e.g., K-omega, k-omega SST, LES. Why did you choose k-epsilon in your study?

Response 9: Thank you so much for your comment. The k-epsilon turbulence model is not outdated, but it is considered a simple turbulence model and has limitations in accurately capturing the behavior of certain flow conditions. The suitability of the k-epsilon turbulence model in simulation depends on the specific application and flow conditions being modelled. It is particularly well suited for simulating flows with moderate turbulence intensity, which is suitable in our cases. When running the simulation in Ansys Fluent, the K-epsilon model has shown that the residual graph is converged and prevents the floating-point error. This model is still widely used in many industrial and environmental applications, where its simplicity and robustness make it a good choice. However, in more complex flow conditions, its predictions may not be as accurate as those obtained from more advanced turbulence models. More advanced turbulence models in Reynolds-averaged Navier-Stokes (RANS) models, such as the k-omega turbulence model or Reynolds Stress Model. The large eddy simulation (LES) model is the most accurate one that can provide more accurate predictions for complex flows. However, these models are also more computationally time-demanding, as shown in Figure 1 below. In summary, a k-epsilon turbulence model is still a useful tool for simulating turbulent flows, but its limitations should be taken into account when considering its use for a specific application such as the epsilon equation contains a term which cannot be calculated at the wall. Therefore, wall functions must be used.

Figure 1: Turbulence Model Available in Fluent.

Point 10: Line 251: Please add a short description about the nodal degree of freedom (DOF) and how it affects the accuracy of the simulation results.

Response 10: Thank you so much for your comment. The short description about the nodal degree of freedom (DOF) and how it affects the accuracy of the simulation results has been added in the manuscript.

Point 11: Line 299: I noticed that in the following articles, an unstructured grid was generated for three different simulations. I suggest the authors to take into account these three papers to emphasis application of unstructured grids in numerical simulations of hydraulic phenomena:

- Flow pressure behavior downstream of ski jumps

- Experimental and numerical study of free-surface flows in a corrugated pipe

- Large eddy simulation of the turbulent flow field around a submerged pile within a scour hole under current condition

Response 11: Thank you so much for your comment. The three papers have been cited in this manuscript.

Point 12: Line 349 to 351: If the characteristics of penstock pipes and butterfly valve are not available, how did you assume and define them to the numerical model? These structures are the most important parts of the hydropower and affect your results.

Response 12: Thank you so much for your comment. The drawing of penstock pipes and butterfly valve are not provided by the dam owner but there is on site observation has been conducted to attained the geometry and dimensions of the penstock pipe and butterfly valve.

Point 13: The legends of Figures 6, 7, 8 and 9 are not legible. Please use high resolution figures.

Response 13: Thank you so much for your comments. The legends of Figure 6,7,8 and 9 has been changed to high resolution figures.

Point 14: Line 388: What is the application of the maximum stress of 73.92 MPa? How do engineer utilize it in design? Please analyze this value in view point of practical application.

Response 14: Thank you so much for your comments. Since the maximum stress of penstock is less than the yield stress of mild steel at 370 MPa, no structural failure will be observed.

Point 15: Line 402: The sentence "This match with the study..." is grammatically incorrect. Please rephrase it.

Response 15: Thank you so much for your comments. The sentence has been removed since the citation are not related to this study.  

Point 16: Please add more details about the “maximum stress” that is the main finding of your study into the ‘Conclusions’.

Response 16: Thank you so much for your comment. The conclusion has been revised.

Reviewer 3 Report

The authors investigated the fluid-structure interaction (FSI) of the bottom outlet structure of Chenderoh Dam by numerical method. The dam system was divided into fluid domain and structure domain, with Computational Fluid Dynamics (CFD) analysis and transient structural analysis respectively adopted to study the two domains, and the analysis results of the two domains were coupled together by system coupling to study the FSI of the dam system. This paper is of significance for the structure safety analysis of hydraulic structure, but there are some problems in the present work. The comments that should be addressed are listed as follows.

1. The reviewer suggests the author to rewrite the abstract rather than extract some sentences from the paper verbatim. The abstract should comprise the object, methods, and findings of this paper explicitly and concisely.

2. What does the abbreviation “FSL” mean in line 55? The authors should add an explanation to the abbreviations in this paper when first using them.

3. Line 156-160, the definition of breaking stress, tension and compression are present here, while they have nothing on the analysis of the numerical results in this paper, which is improperly presented in this paper.

4. Line 161-165, this content was repeated twice in the paper and should be deleted.

5. Line 254, “equation (3.10)” should be changed to “equation (6)”, the authors should carefully check the whole paper to avoid this type of mistake. 

6. Line 317, the two coefficients must be important to the results of the numerical model, which should be presented in this paper, and an explanation for how to determine the value of the two coefficients should be given.

7. It was difficult to identify the information presented in Figure 6 (c) and (d) and Figure 7. These figures are too fuzzy to visualize the contour and data of the corresponding results, please replace them with high-quality figures.

8. Line 387, change “opened” to “closed”. According to the subsequent content, case 2 should correspond to the condition of the butterfly valve under fully closed state.

9. Line 396-402, since the authors compared their results with the studies of (Manafpour & Rovesht, 2017) and (Parsaie et al., 2015), the reviewer suggests the author to add a comparison of test data between the present work and the two previous works.

10. What does the horizontal axis mean in figure 10? It is not clear what the distance illustrated in figure 10 is referred to.

11. Line 411-412, where does the percentage deviation 2.34% come from? There was no comparison of test data between the experiment and the numerical model presented in this paper.

12. The conclusion should be rewritten. The main findings written by the authors after discussing the results of the numerical model should be presented in a Conclusion section.

Author Response

Point 1: The reviewer suggests the author to rewrite the abstract rather than extract some sentences from the paper verbatim. The abstract should comprise the object, methods, and findings of this paper explicitly and concisely.

Response 1: Thank you so much for your comments. The abstract has been amended and rewritten.

Point 2: What does the abbreviation “FSL” mean in line 55? The authors should add an explanation to the abbreviations in this paper when first using them.

Response 2: Thank you so much for your comments. The meaning of abbreviation “FSL” has been included in this manuscript.

Point 3: Line 156-160, the definition of breaking stress, tension and compression are present here, while they have nothing on the analysis of the numerical results in this paper, which is improperly presented in this paper.

Response 3: Thank you so much for your comments. The definition of breaking stress, tension and compression has been removed.

Point 4: Line 161-165, this content was repeated twice in the paper and should be deleted.

Response 4: Thank you so much for your comments. The repeated content in this paper has been removed.

Point 5: Line 254, “equation (3.10)” should be changed to “equation (6)”, the authors should carefully check the whole paper to avoid this type of mistake.

Response 5: Thank you so much for your comments. The “equation (3.10)” has been changed to “equation (7)” and the whole paper has been checked to avoid the same mistake.

Point 6: Line 317, the two coefficients must be important to the results of the numerical model, which should be presented in this paper, and an explanation for how to determine the value of the two coefficients should be given.

Response 6: Thank you so much for your comment. An explanation for how to determine the value of the two coefficients has been added in the manuscript.

Point 7: It was difficult to identify the information presented in Figure 6 (c) and (d) and Figure 7. These figures are too fuzzy to visualize the contour and data of the corresponding results, please replace them with high-quality figures.

Response 7: Thank you so much for your comments. The Figure 6 (c) and (d) and Figure 7 has been changed to high-quality figures.

Point 8: Line 387, change “opened” to “closed”. According to the subsequent content, case 2 should correspond to the condition of the butterfly valve under fully closed state.

Response 8: Thank you so much for your comments. The word “opened” has been changed to “closed” according to the content subsequent.

Point 9: Line 396-402, since the authors compared their results with the studies of (Manafpour & Rovesht, 2017) and (Parsaie et al., 2015), the reviewer suggests the author to add a comparison of test data between the present work and the two previous works.

Response 9: Thank you so much for your comment. The compared results with the studies of (Manafpour & Rovesht, 2017) has been added in the manuscript. The studies of (Parsaie et al., 2015) has been removed since the study conducted at the spillway part.

Point 10: What does the horizontal axis mean in figure 10? It is not clear what the distance illustrated in figure 10 is referred to.

Response 10: Thank you so much for your comment. The distance illustrated in figure 10 is referred to distance from intake to gate where the piezometer is located referring to study conducted by Manafpour & Rovesht, 2017.

Point 11: Line 411-412, where does the percentage deviation 2.34% come from? There was no comparison of test data between the experiment and the numerical model presented in this paper.

Response 11: Thank you so much for your comment. The line 411-412 has been rewritten. There was no comparison of test data between the experiment and the numerical model presented in this manuscript.

Point 12: The conclusion should be rewritten. The main findings written by the authors after discussing the results of the numerical model should be presented in a Conclusion section.

Response 12: Thank you so much for your comments. The conclusion has been rewritten

Round 2

Reviewer 2 Report

The manuscript is acceptable in present form.

Reviewer 3 Report

The present paper has been improved, indicating that all the comments have been well addressed. As a result, the paper can be accepted for publication.